# G protein βγ subunits inhibit TRPM3 ion channels in sensory neurons

**Talisia Quallo\*, Omar Alkhatib, Clive Gentry, David A Andersson, Stuart Bevan\***

Wolfson Centre for Age-Related Diseases, King's College London, London, United Kingdom

**Abstract** Transient receptor potential (TRP) ion channels in peripheral sensory neurons are functionally regulated by hydrolysis of the phosphoinositide PI(4,5)P$_2$ and changes in the level of protein kinase mediated phosphorylation following activation of various G protein coupled receptors. We now show that the activity of TRPM3 expressed in mouse dorsal root ganglion (DRG) neurons is inhibited by agonists of the G$_i$-coupled μ opioid, GABA-B and NPY receptors. These agonist effects are mediated by direct inhibition of TRPM3 by G$\beta\gamma$ subunits, rather than by a canonical cAMP mediated mechanism. The activity of TRPM3 in DRG neurons is also negatively modulated by tonic, constitutive GPCR activity as TRPM3 responses can be potentiated by GPCR inverse agonists. GPCR regulation of TRPM3 is also seen in vivo where G$_{i/o}$ GPCRs agonists inhibited and inverse agonists potentiated TRPM3 mediated nociceptive behavioural responses.

\*For correspondence: talisia. quallo@kcl.ac.uk (TQ); stuart. bevan@kcl.ac.uk (SB)

**Competing interests:** The authors declare that no competing interests exist.

## Introduction

Proteins encoded by the TRPM3 gene form non-selective cation channels which are widely expressed in mammalian tissues. The discovery that TRPM3 can be activated by the endogenous neurosteroid pregnenolone sulphate (PS), has facilitated the study of this widely-expressed TRP channel and PS has been utilised as a pharmacological tool for channel characterisation and as a probe for TRPM3 expression (*Wagner et al., 2008*). TRPM3 is expressed in peripheral sensory neurons where it acts as a heat sensor (*Vriens et al., 2011*). Activation of TRPM3 channels in vivo has been shown to evoke nociceptive behaviours and mice without functional TRPM3 channels exhibit altered temperature preferences, compromised behavioural responses to noxious heat and fail to develop heat hyperalgesia associated with inflammation (*Vriens et al., 2011*).

There have been relatively few studies of the mechanisms which regulate or sensitise TRPM3. Many TRP channels are regulated by signalling pathways associated with activation of G-protein coupled receptors (GPCRs). For example, activation of both Gα$_s$ and Gα$_q$- coupled receptors can sensitise the heat sensitive nociceptor TRPV1 through protein kinase-dependent mechanisms (*Bevan et al., 2014*). Like other TRP channels, TRPM3 can be regulated by phosphoinositol 4,5-bisphosphate (PI(4,5)P$_2$) and other phosphoinositides as loss or hydrolysis of PI(4,5)P$_2$ leads to a reduction in TRPM3 activity that can be restored by application of exogenous PI(4,5)P$_2$ (*Badheka et al., 2015*; *Tóth et al., 2015*). These findings suggest that TRPM3 activity can be regulated downstream of activation of G$_q$ coupled GPCRs. A human TRPM3 variant with a short carboxyl terminus was found to be insensitive to stimulation of G$_q$-coupled muscarinic receptors or histamine H1 receptors (*Grimm et al., 2003*). However, another human splice variant, TRPM3a, was shown to be activated by muscarinic receptor stimulation (*Lee et al., 2003*), suggesting that individual genetic variants can be differentially regulated. Units of the G protein itself can directly interact with channel structures to modulate ion channel activity. Indeed activation of Gα$_q$-linked receptors inhibits TRPM8 via a direct action of the Gα$_q$ subunit with the TRPM8 channel (*Zhang et al., 2012*)

**eLife digest** TRPM3 belongs to a family of channel proteins that allow sodium and calcium ions to enter cells by forming pores in cell membranes. TRPM3 is found on the cell membranes of nerve cells; when ions flow into the nerves through the TRPM3 pores it triggers an electrical impulse. TRPM3 is responsible for helping us to detect heat, and mice without this protein find it difficult to sense painfully hot temperatures. Mice lacking TRPM3 also respond to other kinds of pain differently. Normally, a mouse with an injured paw becomes more sensitive to warm and hot temperatures, but this does not happen in mice that do not have TRPM3.

When activated, other proteins called G-protein coupled receptors (or GPCRs for short) can make some members of this family of channel proteins more or less likely to open their pore. This in turn increases or decreases the flow of ions through the pore, respectively. Yet it was not clear if GPCRs also affect TRPM3 channels on the membranes of nerve cells.

Quallo et al. have now discovered that "switching on" different GPCR proteins in sensory nerve cells from mice greatly reduces the flow of calcium ions though TRPM3 channels. The experiments made use of two pain-killing drugs, namely morphine and baclofen, and a molecule called neuropeptide Y to activate different GPCRs.

GPCRs interact with a group of small proteins called G-proteins that, when activated by the receptor, split into two subunits, known as the $\alpha$ subunit and the $\beta\gamma$ subunit. Once detached these subunits are free to act as messengers and interact with other proteins in the cell membrane. Quallo et al. found that TRPM3 is one of a small group of proteins that interact with the $\beta\gamma$ subunits of the G-protein, which can explain how "switching on" GPCRs reduces the activity of TRPM3. Two independent studies by Dembla, Behrendt et al. and Badheka, Yudin et al. also report similar findings.

There is currently a need to find more effective treatments for people suffering from long-term pain conditions and it has become clear that TRPM3 channels are involved in sensing both pain and temperature. These new findings show that drugs already used in the treatment of pain can dramatically change how TRPM3 works. These results might help scientists to find drugs that work in a similar way to dial down the activity of TRPM3 and to combat pain. Though first it will be important to confirm these new findings in human nerve cells.

and TRPM1 is inhibited by interactions with either $G\alpha$ or $G\beta\gamma$ subunits (*Shen et al., 2012*; *Xu et al., 2016*).

In this study we have examined the effects of GPCR activation on the function of endogenously expressed TRPM3 channels in isolated DRG neurons from mice and investigated if activation of $G_{i/o}$ coupled GPCRs can modulate TRPM3 mediated nociceptive responses in vivo. For these experiments we have studied three different receptors (opioid, $GABA_B$ and neuropeptide Y) which are known to modulate sensory neuron activity (*Levine and Taiwo, 1989*; *Schuler et al., 2001*; *Smith et al., 2007*). We have also examined the underlying mechanism in CHO and HEK293 cells exogenously expressing TRPM3. Our results demonstrate that activation of these GPCRs inhibits TRPM3 activity in DRG neurons by a $G\beta\gamma$ mediated mechanism and inhibits PS-evoked nociceptive responses in mice.

## Results

### Morphine, Baclofen and PYY inhibit TRPM3 mediated PS-induced Ca$^{2+}$ responses

Opioids remain one of the best known and most effective treatments for pain, and $G_{i/o}$ coupled GPCRs for opioid ligands are expressed on sensory neurons where receptor activation inhibits voltage-gated calcium channels (VGCCs) (*Stein et al., 2003*). Therefore, to determine if activation of $G_{i/o}$ GPCRs can modulate natively expressed TRPM3 channels, we first examined the effects of the prototypical opioid receptor agonist, morphine.

We used two consecutive applications of a submaximal concentration of pregnenolone sulphate (PS, 20 µM,) to investigate the effect of morphine on TRPM3-mediated $[Ca^{2+}]_i$-responses in isolated DRG neurons. This concentration of PS typically activates about 30% of cells in DRG cultures. The first application of PS was used to identify TRPM3 expressing neurons and the second to assess the effects of pharmacological treatments. We refer to the response amplitude of the second PS challenge as R (relative % response). In control experiments, the second PS challenge evoked $[Ca^{2+}]_i$-responses (R) that were $63 \pm 2\%$ of the first PS response amplitude (*Figure 1a*). Treatment with morphine (10 µM) for 2 min before and during the second PS challenge significantly reduced R to $12 \pm 1\%$ (p<0.001, *Figure 1b,c*). In the majority of PS-responsive neurons (55%, n = 115/209), application of morphine completely abolished PS-evoked $[Ca^{2+}]_i$ responses (R < 5%).

To confirm that the inhibitory effect of morphine is receptor mediated and not a direct effect of morphine on TRPM3, we examined the effect of morphine in the presence of the opioid receptor antagonist, naloxone. Naloxone (1 µM) inhibited the effect of morphine (10 µM) completely, since the relative amplitude (R) produced by the second PS challenge in the presence of morphine plus naloxone was $73 \pm 3\%$ (p>0.05, *Figure 1d*), very similar to untreated control neurons (R = $64 \pm 3\%$). Thus morphine inhibits TRPM3 by activating an opioid receptor present on DRG neurons.

The involvement of $G_{i/o}$ proteins was investigated using pertussis toxin (PTX), which inhibits the coupling of $G_{i/o}$ proteins to their cognate GPCRs by catalysing ADP-ribosylation of the $G\alpha_{i/o}$ subunits. Incubation with PTX (200 ng/ml, for ~2 ½ −18 hr) significantly reduced the inhibitory effect of morphine in a large proportion of the neurons (morphine: R = $16 \pm 2\%$; morphine +PTX: R = $46 \pm 2\%$, p<0.001, *Figure 1d*). These findings indicate that activation of a sensory-neuron expressed opioid receptor and subsequent $G\alpha_{i/o}$ protein signalling is able to modulate the activity of TRPM3 channels. Opioid receptors, like many other types of GPCRs, may possess constitutive, agonist independent activity (*Rosenbaum et al., 2009*). We tested whether such tonic receptor activity modulates the activity of endogenously expressed TRPM3 in DRG neurons, using two consecutive challenges of a low concentration of PS (5 µM) that evoked $[Ca^{2+}]_i$-responses in only a small percentage of neurons. In experiments where cells were exposed to naloxone during the second PS challenge the number of responding neurons increased significantly (p<0.001, Fisher's) from 3.6% (n = 36/999) to 9% (90/999, *Figure 1e*). In contrast, the number of PS responders fell from 3.5% (n = 40/1153) to 1.8% (n = 21/1153) in control experiments. These findings indicate that constitutive activity of opioid receptors exerts a tonic inhibition of TRPM3 in DRG neurons.

We used patch-clamp recordings to confirm that the observed opioid receptor mediated inhibition of PS-evoked $[Ca^{2+}]_i$-responses is associated with a corresponding inhibition of TRPM3 currents in DRG neurons. TRPM3 currents have a marked outward rectification (*Wagner et al., 2008*) and we therefore studied neurons at a holding potential of +40 mV. Application of PS (50 µM) rapidly evoked membrane currents in a subset of isolated DRG neurons (*Figure 1f*). Subsequent introduction of morphine (10 µM) in the continued presence of PS produced a rapid, near-complete and reversible inhibition of the PS-evoked current. We next examined the effect of morphine on PS-evoked currents in CHO cells co-expressing TRPM3 and the µ opioid receptor. PS evoked large currents that were reversibly inhibited by co-application of morphine (inhibition $89 \pm 4\%$, n = 9). Examination of the current-voltage relationships before and during PS application and in the presence of morphine (*Figure 1g*) showed that the response to PS was greatly inhibited by morphine at all voltages studied.

Morphine is a non-selective opioid receptor agonist and DRG neurons express all three naloxone-sensitive opioid receptor subtypes (µ, δ and κ). We examined the effects of sub-type selective opioid receptor agonists on PS-evoked $[Ca^{2+}]_i$ responses of DRG neurons, to determine which receptor subtype is important for morphine-induced inhibition of TRPM3. Treatment with the selective κ opioid receptor agonist U50488 (20 nM), did not inhibit PS-induced responses and instead produced a modest, but significant increase (U50488: R = $72 \pm 3\%$; control: R = $57 \pm 3\%$, p<0.05, *Figure 2a*). The δ opioid receptor agonist SB205607 (20 nM) was without effect (SB205607: R = $52 \pm 3\%$; p>0.05, *Figure 2a*), whereas the µ opioid receptor agonist DAMGO (20 nM) significantly inhibited PS-evoked $[Ca^{2+}]_i$-responses to an extent similar to that previously observed with morphine (DAMGO: R = $18 \pm 2\%$, p<0.001, *Figure 2a,b*). These findings indicate that morphine predominantly inhibits TRPM3 by activating µ opioid receptors and suggest a high degree of co-expression of µ opioid receptors and TRPM3. These results are in keeping with earlier observations of µ (but not δ) opioid receptor expression in heat nociceptors (*Scherrer et al., 2009*).

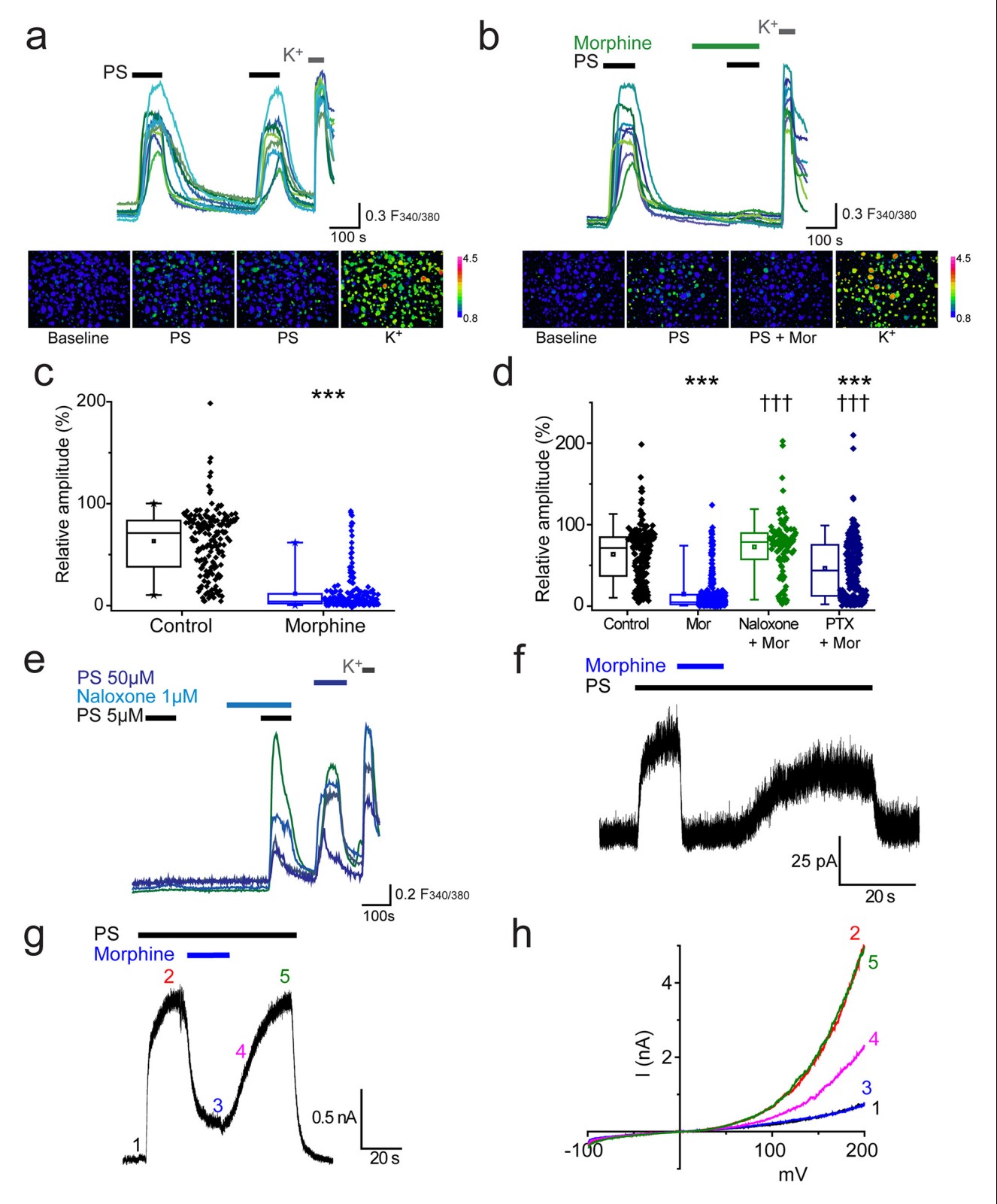

**Figure 1.** Morphine inhibits TRPM3 channels expressed on sensory neurons. (**a**) Traces of DRG $[Ca^{2+}]_i$ responses evoked by two sequential PS challenges (20 µM) followed by high $K^+$ (50 mM KCl). (**b**) Effect of treatment with 10 µM morphine for 2 min before and during the second PS challenge. $F_{(340/380)}$ indicates fura-2 emission ratio. The lower panels in (**a**) and (**b**) are pseudocolour images illustrating the change in F340/380 ratio in response to PS and KCl. The bar indicates the colours corresponding to various F340/380 values. (**c**) Box and whisker plots and data points showing the amplitudes

*Figure 1 continued on next page*

*Figure 1 continued*

for responses to the second PS challenge in (a) and (b), ***p<0.001; Mann-Whitney U test (control, n = 174; morphine, n = 209). (d) Effect of treatment with morphine (10 µM, n = 323), morphine (10 µM) and naloxone (1 µM, n = 110) and morphine (10 µM) following an incubation with pertussis toxin (200 ng/ml for 2.5 h-18h, n = 253) on $[Ca^{2+}]_i$ responses evoked by the second PS (20 µM) challenge using the protocol in (a and b). Control group, n = 188. (e) Traces displaying neuronal $[Ca^{2+}]_i$ responses to two PS (5 µM) challenges in the absence and presence of naloxone (1 µM), followed by a 50 µM PS challenge and high K+ (50 mM KCl). ***p<0.001, compared to control. ††† p<0.001, compared to morphine (10 µM), Kruskal-Wallis. (f) Whole cell recording illustrating that morphine reversibly inhibits PS-evoked outward membrane currents in DRG neurons (+40 mV). (g) Inhibitory effect of morphine on whole cell outward current in a CHO cell co-expressing TRPM3 and µ-opioid receptor (+60 mV). (h) Current-voltage relationships for another TRPM3/µ-opioid receptor expressing CHO cell measured at times corresponding to time points 1–5 in panel g.

We next examined whether GPCR inhibition of TRPM3 was specific to opioid receptors or could be extended to other $G_{i/o}$-coupled receptors expressed on sensory neurons. The metabotropic $G_{i/o}$-coupled receptors for GABA, GABA$_{B1}$ and GABA$_{B2}$, are expressed by a high percentage (60–90%) of peripheral sensory neurons (*Charles et al., 2001*; *Cuny et al., 2012*; *Engle et al., 2012*) and

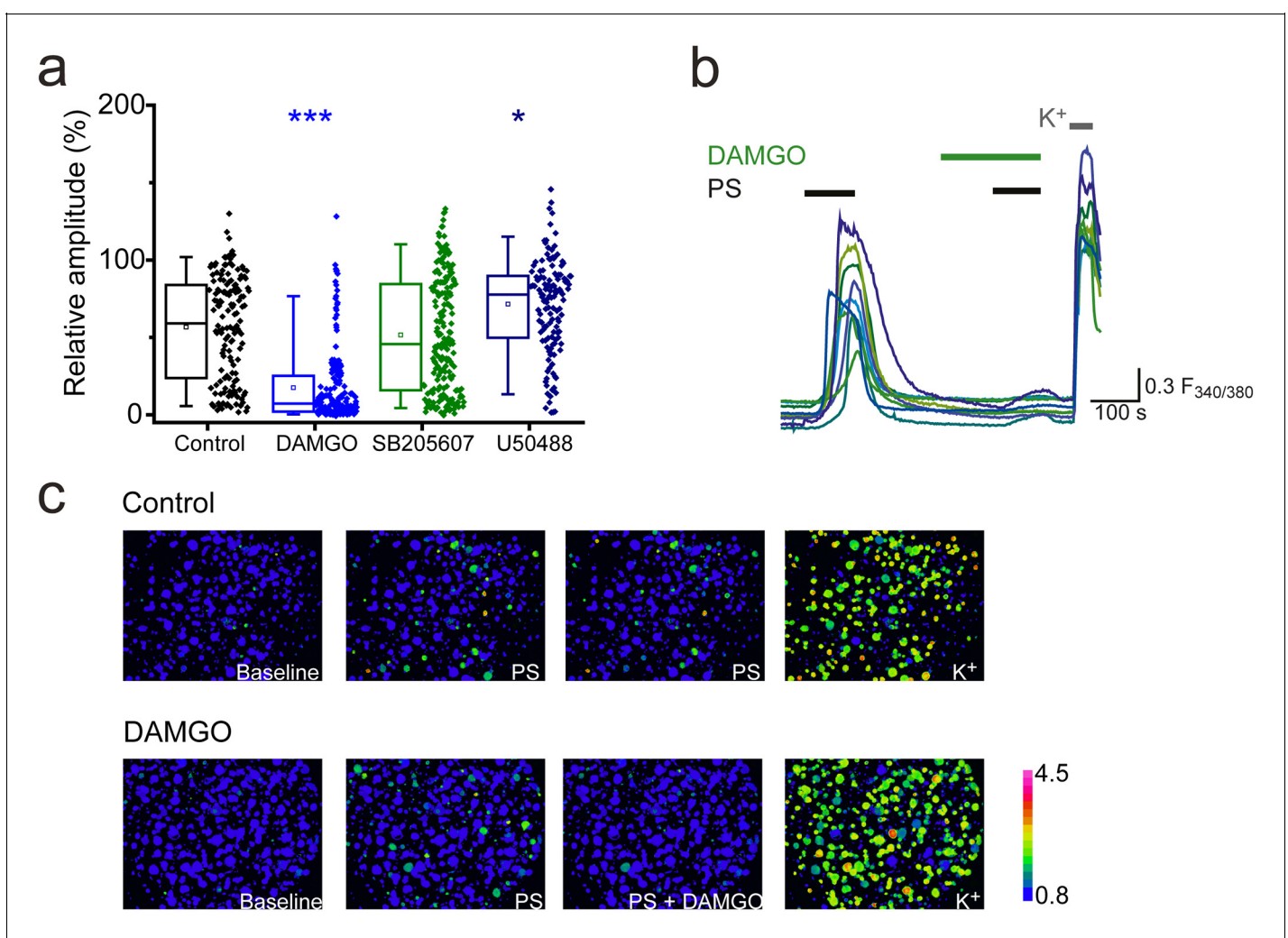

**Figure 2.** Activation of µ-opioid receptors inhibits TRPM3. (a) Effect of the selective µ opioid receptor agonist DAMGO, δ opioid receptor agonist SB205607 and κ opioid receptor agonist U50488 (each at 20 nM) on $[Ca^{2+}]_i$-responses evoked by PS (20 µM, second application, see *Figure 1* for protocol). *p<0.05, ***p<0.001; Kruskal-Wallis (control, n = 152; DAMGO, n = 184; SB205607, n = 200; U50488, n = 140). (b) Traces displaying the effect of DAMGO (20 nM) on neuronal $[Ca^{2+}]_i$ responses to stimulation with PS (20 µM). $F_{(340/380)}$ indicates fura-2 emission ratio. (c) Pseudocolour images illustrating the change in F340/380 ratio in response to PS and KCl with the colours corresponding to various F340/380 values indicated by the bar.

recently, activation of $GABA_{B1}$ was shown to modulate activity of TRPV1 channels (*Hanack et al., 2015*). Treatment of DRG neurons with the selective $GABA_B$ agonist baclofen (100 µM), significantly reduced the amplitude of the second PS response to R = 12 ± 2% compared to R = 58 ± 6% in control experiments (p<0.001, *Figure 3a,b*) and abolished (R < 5%) the PS-evoked $[Ca^{2+}]_i$ responses in 67% (n = 130/194) of neurons.

The $G_{i/o}$-coupled neuropeptide tyrosine (NPY) receptors Y1 and Y2, are each expressed on 15–20% of sensory neurons (*Zhang et al., 1994*, *Zhang et al., 1997*; *Brumovsky et al., 2005*; *Ji et al., 1994*; *Taylor et al., 2014*). The Y1 receptor is predominantly expressed in small diameter neurons whereas the Y2 receptor is largely expressed in medium and large diameter neurons. We examined whether activation of NPY receptors by the agonist peptide YY (PYY) was able to modulate neuronal TRPM3 $[Ca^{2+}]_i$-responses. Similarly to the effects of morphine and baclofen, application of 100 nM PYY reduced the PS-evoked $[Ca^{2+}]_i$ responses, and in 57% (n = 123/217) of neurons abolished the evoked increase in $[Ca^{2+}]_i$ (R < 5%). The relative amplitude of the second PS-evoked $[Ca^{2+}]_i$ responses was reduced from 66 ± 3% in control experiments to 11 ± 1% in the presence of PYY (p<0.001). Like opioid receptors, Y2 receptors have been reported to possess constitutive activity (*Chen et al., 2000*) and we therefore examined whether tonic Y2 receptor activity modulates TRPM3. In experiments where we challenged neurons with two consecutive applications of a low concentration of PS (5 µM), treatment with the selective Y2 receptor antagonist BIIE 0246 (10µM) significantly (p<0.05, Fisher's) increased the number of responding neurons from 1.1% (n = 5/450) to 4.2% (19/450; *Figure 3e*). In contrast, the number of PS responders fell from 3.4% (n = 15/440) to 1.8% (n = 8/440) in control experiments with two applications of 5 µM PS. These findings indicate that constitutive activity of Y2 receptors can exert a tonic inhibition of TRPM3.

To examine whether opioids, baclofen and PYY target distinct populations of TRPM3-expressing neurons, neurons were sequentially exposed to morphine (10 µM), baclofen (100 µM) and PYY (100 nM) and challenged with PS (20 µM) in the presence and absence of these drugs (*Figure 3f–h*). A large proportion of PS-sensitive neurons (43%, n = 91/210) were inhibited (response amplitude <15% of first PS response amplitude) by all three agonists. A population of neurons was inhibited by morphine and baclofen but not PYY (7%, n = 14/210) and a slightly larger percentage of neurons was inhibited by morphine and PYY but not baclofen (13%, n = 27/210). Moreover, 10% (n = 21/210) of neurons were inhibited by baclofen and PYY but not morphine. Small populations of neurons were inhibited by just one agonist (morphine, 7%, n = 15/210; baclofen, 4%, n = 9/210; PYY, 4%, n = 9/210) and a subpopulation of neurons were not inhibited by any of the agonists (15%, n = 31/210). These findings, which are illustrated schematically in the Venn diagram (*Figure 3i*), indicate that morphine, baclofen and PYY target overlapping populations of TRPM3-expressing neurons.

Additional experiments demonstrated that not all $G_{i/o}$-coupled receptors are able to effectively modulate TRPM3 activity. L-AP4 (5 µM) activation of group III metabotropic glutamate receptors ($mGLUR_{4/6/7/8}$), which are expressed by DRG neurons (*Carlton and Hargett, 2007*; *Govea et al., 2012*), had no significant effect on PS-evoked $[Ca^{2+}]_i$ responses (R = 69 ± 3%, n = 100) compared to control (R = 61 ± 7%, n = 20; p>0.05, Mann-Whitney U test). Cannabinoid CB1 receptors are expressed by a subset of DRG neurons (*Agarwal et al., 2007*; *Veress et al., 2013*). The CB1 receptor agonist WIN 55212–2 (1 µM) slightly but significantly reduced the amplitude of PS evoked $[Ca^{2+}]_i$ responses (R = 51 ± 3%; *Figure 4a*) compared to control (R = 62 ± 2%; p<0.001, Mann-Whitney U test). This finding suggested that application of WIN 55212–2 inhibited PS-evoked responses in some but not in all DRG neurons. This was confirmed when DRG neurons were challenged with 3 applications of 20 µM PS with 1 µM WIN 552212–2 present before and during the second application and 1 µM WIN 55212–2 plus 0.5 µM AM251 (a CB1 receptor antagonist) present during the third PS application (*Figure 4b*). This experimental protocol allowed for detection of neurons whose PS evoked responses were inhibited by WIN 55212–2, and were restored when a CB1 antagonist (AM251) was co-applied with WIN 55212–2. In many DRG neurons, WIN 55212–2 had no obvious inhibitory effect (*Figure 4b*, top), but a small percentage of neurons (5%, n = 8/156) showed an AM251-reversible WIN 552212–2 inhibition of the PS response (*Figure 4b*, bottom). In view of the minor effect of CB1 receptor activation on DRG TRPM3 PS responses, we repeated the experiments in TRPM3 CHO cells transiently transfected with the CB1 receptor. As not all cells expressed CB1 after transfection, we compared the population responses to two consecutive PS applications. PS responses to the two PS applications were relatively stable in control cells not transfected with CB1, whereas WIN 552212–2 inhibited the PS responses in a sub-population of CB1 transfected cells.

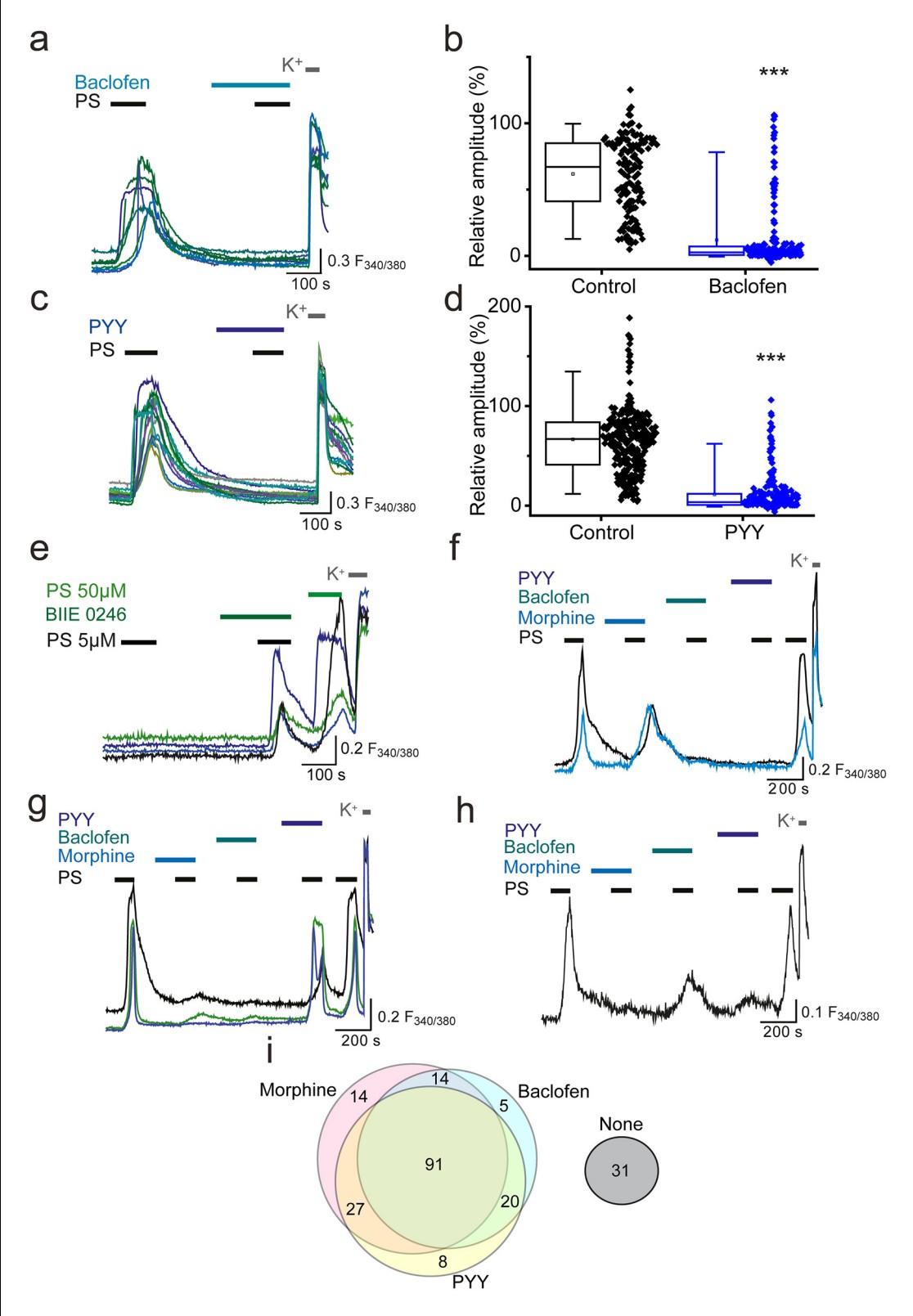

**Figure 3.** Activation of other Gi-coupled receptors also inhibit TRPM3. (**a, b**) Effect of (RS)-Baclofen (100 μM) on the relative [Ca$^{2+}$]$_i$-response amplitude evoked by PS (20 μM). (**c, d**) Effect of PYY (100 nM) on the relative [Ca$^{2+}$]$_i$-response amplitude evoked by PS (20 μM). Plots illustrate the response amplitudes evoked by the second PS challenge (% of first PS amplitude) in control experiments and experiments where neurons were perfused with (**b**)

*Figure 3 continued on next page*

*Figure 3 continued*

(RS)-Baclofen (control, n = 138; Baclofen, n = 194) and (**d**) PYY (control, n = 289; PYY, n = 217). \*\*\*p<0.001; Mann-Whitney U test (**e**) Traces displaying $[Ca^{2+}]_i$ responses to two sequential 5 µM PS challenges, followed by a 50 µM PS challenge and depolarisation with high K⁺ (50 mM KCl). Cells were exposed to 10 µM BIIE 0246 for 2 min before and during the second PS challenge. (**f–h**) Traces displaying DRG $[Ca^{2+}]_i$ responses to sequential 20 µM PS challenges (indicated by black bars) in the presence and absence of morphine (10 µM), baclofen (100 µM) and PYY (100 nM), followed by depolarisation with high K⁺ (50 mM KCl). A group of cells were inhibited by some GPCR agonists but not others. $F_{(340/380)}$ indicates fura-2 emission ratio. (**i**) Venn diagram illustrating the pattern of inhibition of PS responses by morphine, baclofen and PYY, n = 210 PS and KCl responsive cells. Responses in 31 cells were not inhibited by any of the GPCR agonists (second response amplitude >15% of first PS response amplitude).

Furthermore, the second PS response of CB1 transfected cells was augmented by AM251 (*Figure 4c,d*). These results suggest that CB1 receptors can regulate TRPM3 but that there is either little co-expression of TRPM3 and CB1 in DRG neurons or ineffective coupling.

We next investigated whether morphine generally affected TRP channels in DRG neurons by examining its effect on TRPV1 responses to capsaicin. As capsaicin responses readily desensitize, we performed the experiments in the presence of cyclosporin (1 µM) which reduces TRPV1 desensitization (*Docherty et al., 1996*). DRG neurons were stimulated two or three times with 30 nM capsaicin with morphine present before and during the second capsaicin application (*Figure 5a*). Morphine had no significant inhibitory effect using this protocol (*Figure 5b*) in contrast to the marked inhibition of PS responses seen in the same experiments (*Figure 5c*).

## Inhibition of TRPM3 is independent of cAMP and does not rely on $G\alpha_i$ proteins

Activation of $G\alpha_i$ subunits leads to inhibition of adenylate cyclase, the enzyme responsible for producing cAMP (cyclic 3', 5'-adenosine monophosphate). Activation of protein kinase A (PKA) by cAMP and the resulting regulation of ion channel functions by PKA are well characterised. In order to examine whether opioid mediated inhibition of TRPM3 is driven by reduced levels of cAMP, we investigated whether morphine could still exert its inhibitory effects in the presence of a membrane permeable cAMP analogue, 8-bromo cAMP. 8-bromo cAMP was unable to compensate for the morphine-induced inhibition of PS responses. In experiments where 8-bromo cAMP (1 mM) was co-administered with morphine (10 µM) the response amplitude evoked by the second PS challenge was 11 ± 2% (*Figure 6a,b*) which was similar to the amplitude in the presence of morphine alone (13 ± 2%, p>0.05, Kruskal-Wallis).

In addition to initiating second messenger signalling pathways, $G\alpha$ subunits can have direct actions on ion channels. To test the involvement of $G\alpha_i$ subunits in opioid-mediated inhibition of TRPM3 we examined the effect of the selective $G\alpha_i$-inhibitor NF023 (*Freissmuth et al., 1996*) on morphine-induced inhibition of PS responses. Although NF023 has been reported to inhibit $G\alpha_i$-meditated signalling when applied extracellularly to cells (*Sarwar et al., 2015*), this charged molecule probably has limited membrane permeability. We therefore examined electrophysiologically whether intracellularly applied NF023 would modify the effect of morphine on PS evoked currents in TRPM3/µ opioid receptor expressing CHO cells (*Figure 6c*). Inclusion of a high concentration (100 µM) of NF023 in the patch pipette had no marked effect on the inhibitory actions of morphine (*Figure 6d,e*). These findings indicate a $G\alpha_i$- and cAMP-independent mechanism for opioid-induced inhibition of TRPM3.

## Beta-gamma subunits of Gi proteins mediate inhibition of TRPM3

Activation of GPCRs also leads to the release of $G\beta\gamma$ subunits which are effector molecules themselves and can bind directly to ion channels to modulate their function (*Smrcka, 2008*). We therefore examined the effect of morphine and baclofen on PS evoked responses in the presence of a $G\beta\gamma$ inhibitor, gallein. Morphine induced inhibition of PS evoked $\{Ca^{2+}\}_i$-responses was dramatically supressed by gallein (20 µM, *Figure 7a,b*). The response amplitude evoked by the second PS challenge was 109 ± 5% in the presence of both morphine and gallein compared to just 10 ± 1% in the

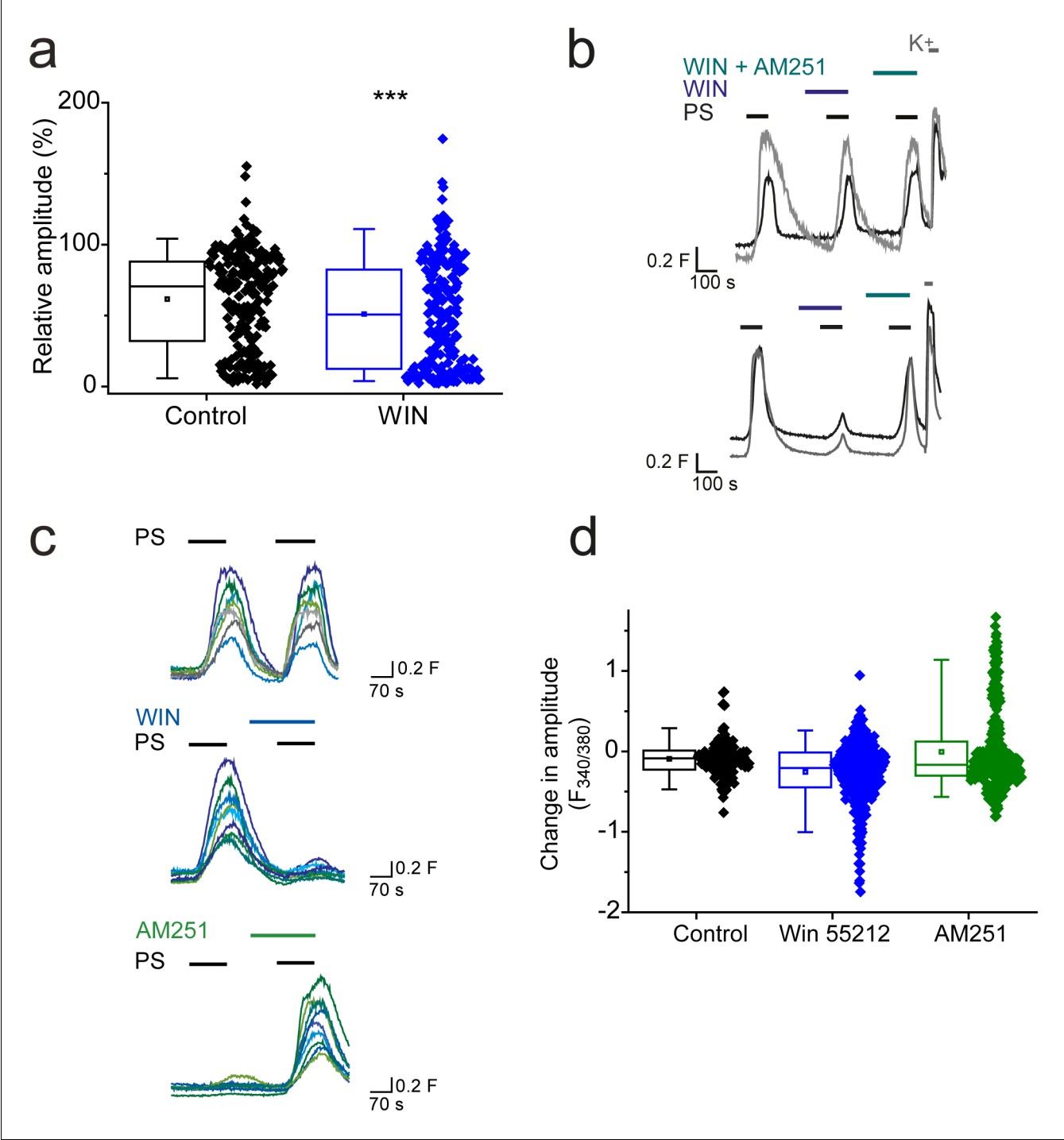

**Figure 4.** CB1 activation can inhibit TRPM3 channels. (a) Effect of CB1 receptor agonist WIN 55212–2 (1 μM) on DRG $[Ca^{2+}]_i$-responses evoked by PS (20 μM, second application). ***p<0.001; Mann-Whitney U test (control, n = 213; WIN 55212–2, n = 218). (b) Traces showing DRG $[Ca^{2+}]_i$ responses to three PS (20 μM) challenges in the absence and presence of WIN 55212–2 (1 μM) and WIN 55212–2 (1 μM) plus AM251 (0.5 μM) followed by high K+ (50 mM KCl). Many PS responses were unaffected by WIN 55212–2 (upper panel) and some showed an inhibition that was reversed by co-application of the antagonist AM251 with WIN 55212–2 (bottom panel). (c) Traces displaying $[Ca^{2+}]_i$ responses to sequential 20 μM PS challenges (indicated by black bars) in CHO cells co-expressing TRPM3 and CB1 receptors in the absence (top) and presence (middle) of 1 μM WIN 552212–2 during the second PS challenge. Middle traces show cells where WIN 552212–2 inhibited the PS responses. Bottom: $[Ca^{2+}]_i$ responses to sequential applications of 5 μM PS with 0.5 μM present during the second PS application. (d) Plots showing the change in $[Ca^{2+}]_i$ response amplitudes (second – first response). Note the increased number of negative values (inhibition) and the increased number of positive values (potentiation) in the presence of the agonist WIN 552212–2 and antagonist AM251, respectively.

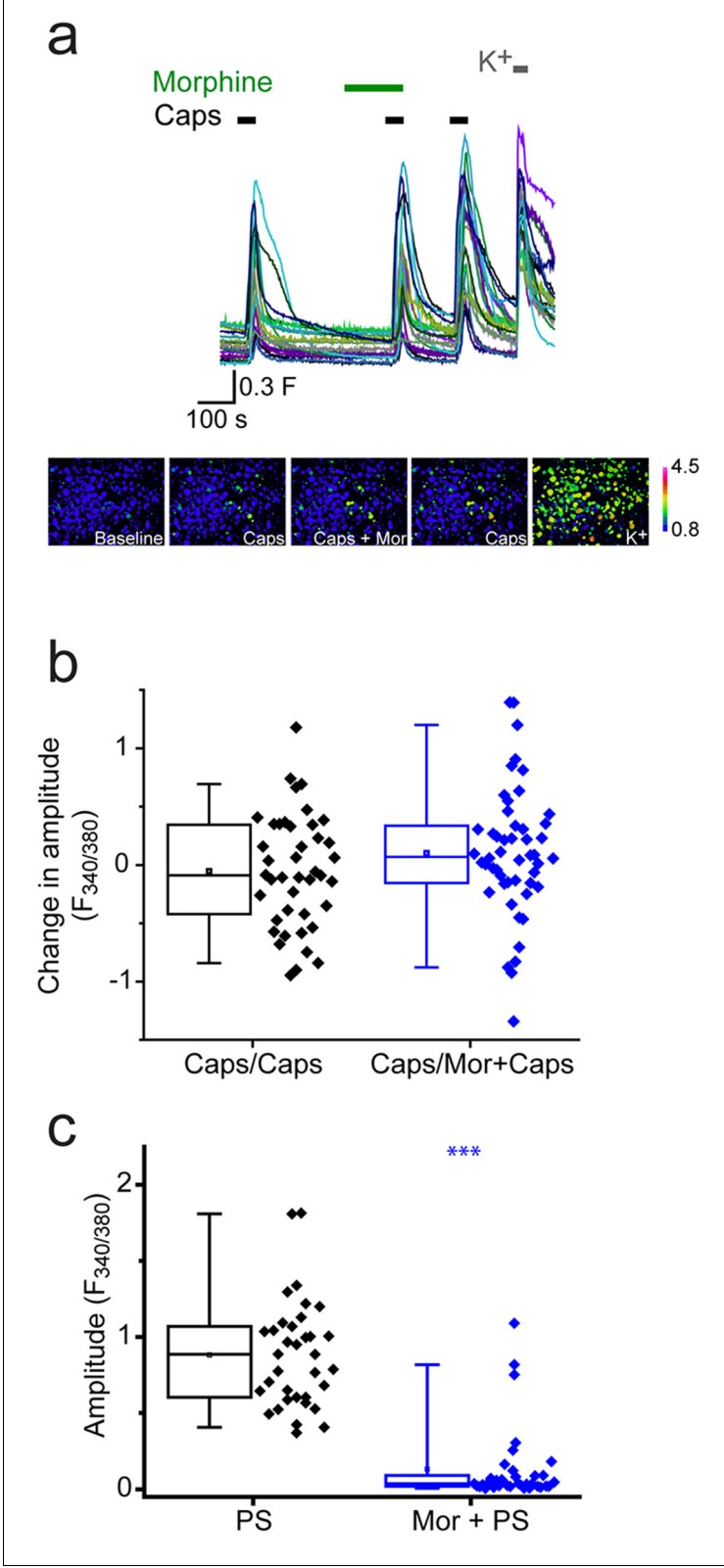

**Figure 5.** Morphine does not inhibit capsaicin evoked $[Ca^{2+}]_i$-responses. Cyclosporin (1μM) included in all solutions to reduce TRPV1 desensitization. (a) Traces showing DRG $[Ca^{2+}]_i$ responses to three capsaicin (1 μM) challenges (indicated by black bars) followed by high $K^+$ (50 mM KCl). Morphine (10 μM was present) before and during the second capsaicin challenge. Lower panels are pseudocolour images illustrating the change in F340/380

*Figure 5 continued on next page*

*Figure 5 continued*

ratio in response to capsaicin and KCl. (**b**) Plots showing the change in $[Ca^{2+}]_i$ response amplitudes (second – first response) under control conditions (left, Caps/Caps) and when morphine was applied (middle, Caps/Mor + Caps). (**c**) 10 µM morphine reduced $[Ca^{2+}]_i$ responses evoked by 20 µM PS in same experiment. Amplitudes of responses evoked by first PS application (left) and second application of PS in presence of morphine (right). ***p<0.001

presence of morphine alone. In additional experiments with three PS challenges, a group of neurons were identified which showed reduced responses (<30% of first response) to PS when morphine (10 µM) was present but regained sensitivity (>50% of first response) when gallein (20 µM) was also present (*Figure 7c,d*).

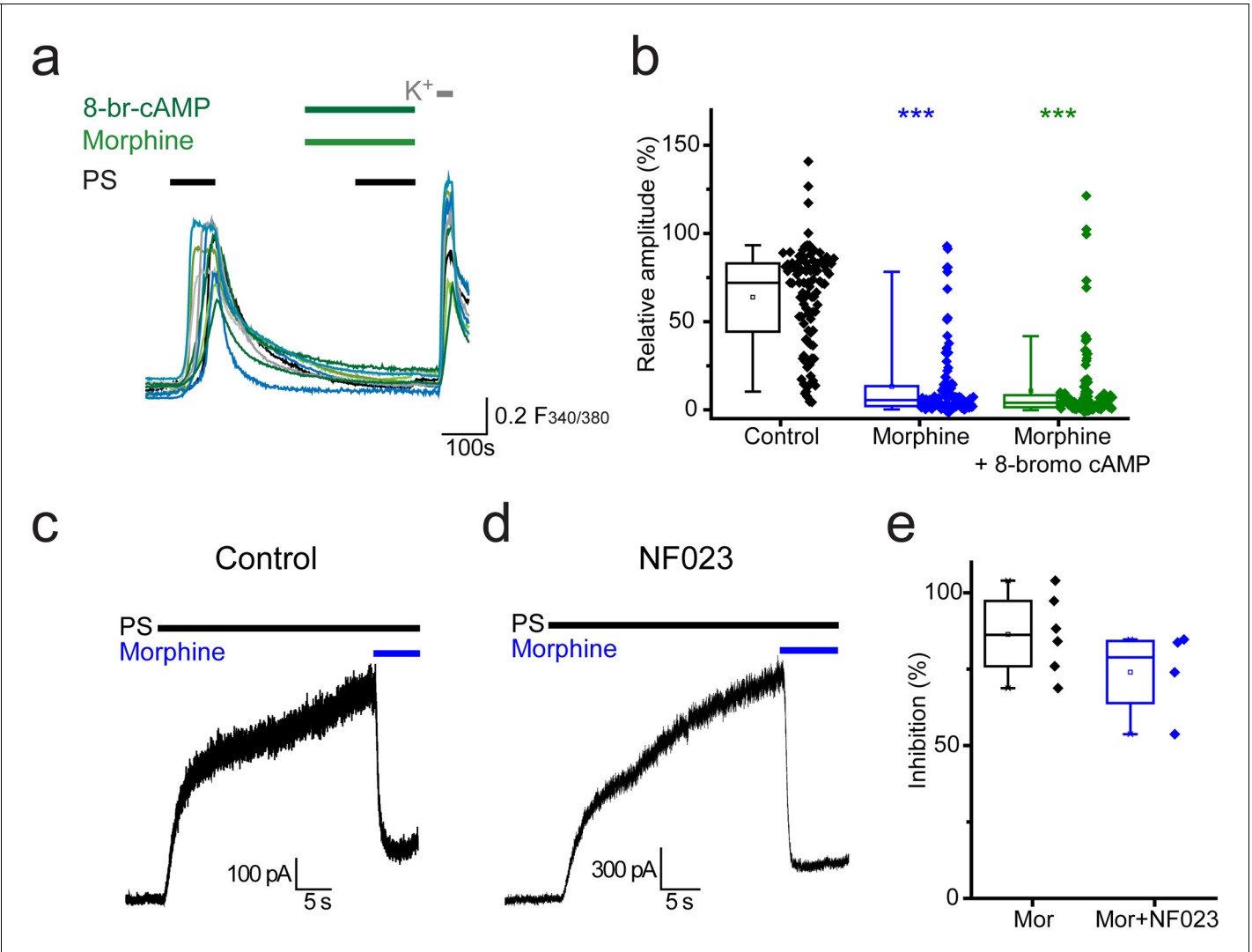

**Figure 6.** Opioid-mediated inhibition of TRPM3 is independent of cAMP and $G\alpha_i$ subunits. (**a**) Traces showing the effect of 8-bromo cAMP (1 mM) on morphine (10 µM) induced inhibition of $[Ca^{2+}]_i$–responses evoked by PS (20 µM) in DRG neurons. $F_{(340/380)}$ indicates fura-2 emission ratio. (**b**) Box and whisker and scatter plots displaying the average response amplitudes from experiments such as **a**., control, n = 96; morphine, n = 95; morphine +8 bromo cAMP, n = 118) ***p<0.001, compared to control. (**c**) Whole cell recordings illustrating morphine inhibition of PS-evoked outward membrane currents in CHO cells co-expressing TRPM3 and the µ opioid receptor (+60 mV) in the absence (left) and presence (right) of 100 µM NF023 in the pipette solution. (**d**) Box and whisker and scatter plots showing the percentage inhibition of 50 µM PS-evoked currents in the absence and presence of NF023. No significant difference noted (p=0.1).

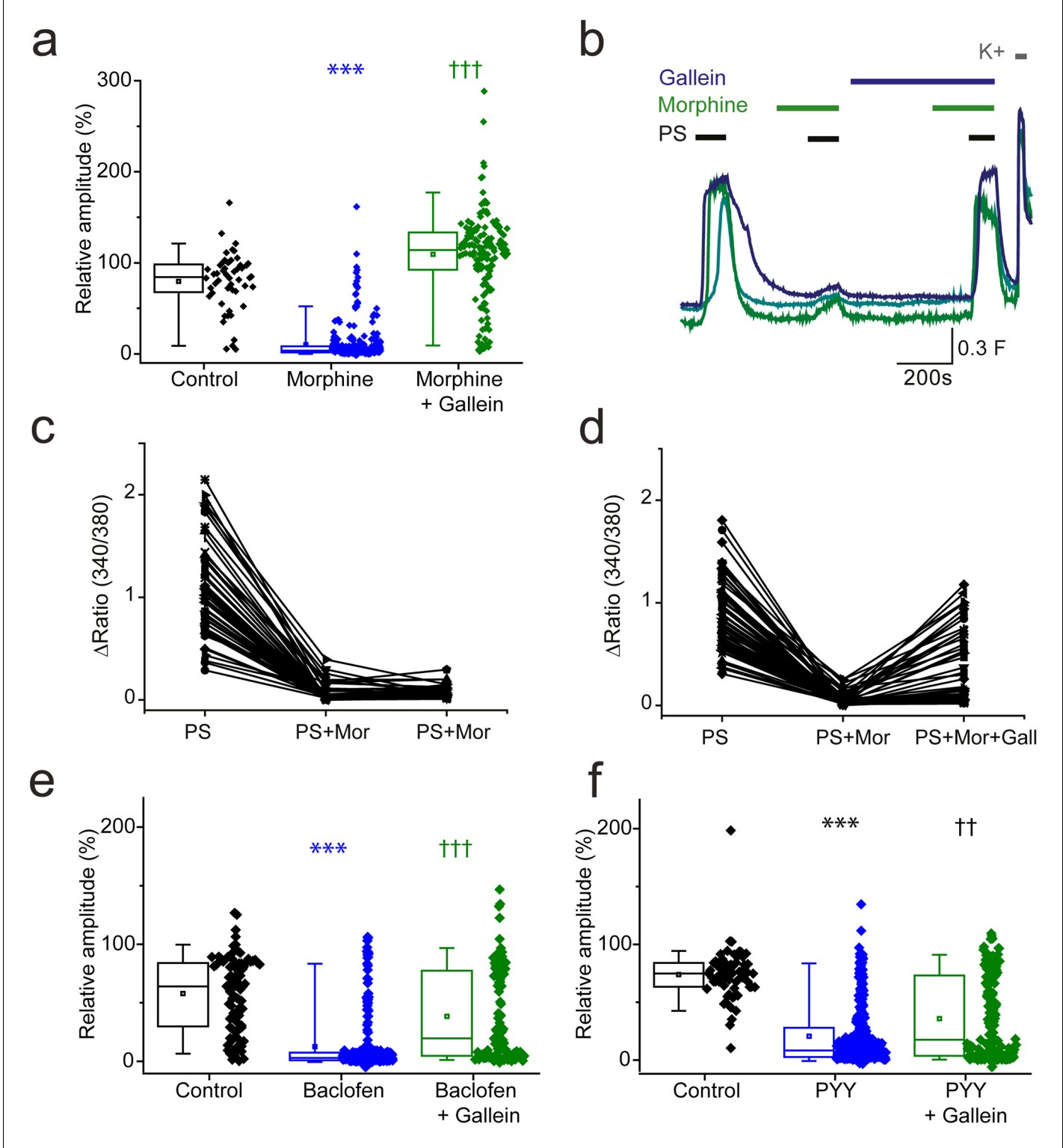

**Figure 7.** $\beta\gamma$ subunits mediate $G_{i/o}$ inhibition of TRPM3. (a, b) Effect of the $G\beta\gamma$ inhibitor gallein (20 µM) on morphine (10 µM) induced inhibition of PS-evoked $[Ca^{2+}]$i-responses. (a) Plots show the relative response amplitudes evoked by a second PS challenge (20 µM) in control conditions, in the presence of morphine and gallein, control n = 52; morphine n = 314; morphine + gallein n = 153. (b) Representative traces demonstrating that gallein reverses morphine inhibition of PS (20 µM). Cells were perfused with 10 µM morphine 2 min before and during the second and third PS challenge, and perfused with 20 µM gallein 7 min before and during the third PS challenge. The results shown are for morphine sensitive cells (PS response amplitude in presence of morphine < 30% of the first response amplitude). ***p<0.001 compared to control, †††p<0.001 compared to morphine, Kruskal Wallis.

*Figure 7 continued on next page*

Figure 7 continued

(c) Scatter and line plot showing the relationship between maximum response amplitudes (Δ Fura-2 ratio) for the first, second and third PS responses in control experiments without gallein and (d) in experiments with gallein (20 µM) present during the third PS application. (e,f) Plots displaying the relative response amplitudes evoked by the second (20 µM) PS challenge for control experiments and experiments where neurons were perfused with (e) 100 µM baclofen or 10 µM gallein and 100 µM baclofen (control, n = 88 baclofen, n = 236; baclofen and gallein, n = 102). ***p<0.001 compared to control, †††p<0.001 compared to baclofen, Kruskal Wallis. (f) Effect of 100 nM PYY or 10 µM gallein and 100 nM PYY on the relative amplitude of PS evoked [Ca$^{2+}$]$_i$-responses (control, n = 65; PYY, n = 288; PYY and gallein, n = 183). ***p<0.001 compared to control, ††p=0.01 compared to PYY. Kruskal-Wallis. F$_{(340/380)}$ indicates fura-2 emission ratio.

We next examined whether gallein also prevents baclofen-induced inhibition of PS-evoked [Ca$^{2+}$]$_i$-responses. When both gallein (10 µM) and baclofen (100 µM) were present during the second PS application, the maximum response amplitude was significantly higher than when baclofen was present alone (gallein and baclofen: 38 ± 4%, baclofen: 12 ± 2%, p<0.001, (*Figure 7e,f*). Similarly, the inhibitory effect of PYY (100 nM) on PS-evoked [Ca$^{2+}$]$_i$-responses was attenuated in the presence of gallein (20 µM); the maximum response amplitude in the presence of both gallein and PYY was significantly higher than the maximum response amplitude in the presence of PYY alone (gallein and PYY: 36 ± 3%, PYY: 21 ± 2%, p<0.001). Intriguingly, the scatter plots suggest that the inhibitory effect of gallein on baclofen and PYY mediated inhibition only occurred in a subset of neurons, unlike the strong effect noted for morphine inhibition in almost all cells. Nevertheless, these results indicate that TRPM3 activity can be regulated by a βγ-dependent mechanism after activation of opioid, GABA$_B$ and NPY receptors.

To determine whether TRPM3 can be inhibited directly by Gβγ subunits in the absence of GPCR stimulation and Gα$_i$ activation, we examined electrophysiologically the effect of intracellularly dialysing TRPM3 expressing HEK293 cells with Gβγ subunits (50 nM). Application of PS (50 µM) for 10 s every minute evoked large outward currents in TRPM3 HEK293 cells, without any detectable desensitization (*Figure 8a,b*). In contrast, intracellular dialysis with βγ-subunits (50 nM) produced a significant and progressive inhibition of the PS-evoked current amplitude (*Figure 8a,b*). To confirm the inhibitory effects of Gβγ subunits on TRPM3 we applied Gβγ subunits to the intracellular face of inside-out membrane patches from TRPM3 HEK293 cells stimulated with another, specific TRPM3 agonist, CIM0216. Macroscopic CIM-0216-evoked currents in these patches were greatly inhibited by 50 nM Gβγ subunits (Fig, 8 c, e) but not by administration of Gβγ subunits that had been heat denatured by boiling (*Figure 8d,e*). These results are consistent with direct inhibition of TRPM3 by Gβγ subunits.

## G$_{i/o}$ coupled GPCRs modulate TRPM3 mediated nociceptive responses

We determined the effects of TRPM3 modulation in vivo by examining the nociceptive responses evoked by TRPM3 agonists in wild-type mice. Previous studies in TRPM3 knockout mice have shown that the behavioural responses to intraplantar injection of PS and CIM0216, are dependent on TRPM3 (35). The behavioural responses to either agonist alone were often mild. We therefore administered a combination of 5 nmole PS and 0.5 nmole CIM-0216 as these compounds have been shown to act synergistically in vitro (*Held et al., 2015*). Intraplantar administration of the combined agonists evoked robust paw licking/flinching behaviour that was measured over a 2 min period. Prior intraplantar administration of morphine (130nmole) essentially abolished the behavioural response evoked by PS/CIM0216 (*Figure 9a*), and intraplantar injections of either baclofen (240nmole) or PYY (235pmole) also significantly reduced the response (*Figure 9b*). Behavioural responses could be inhibited by morphine acting on, for example, voltage gated potassium channels in the periphery. We therefore examined the effects of intraplantar morphine on the paw licking/flinching responses evoked by intraplantar injection of capsaicin. Morphine had no inhibitory effect on capsaicin-elicited behaviours (*Figure 9e*) in marked contrast to its effect on PS/CIM0216 responses.

Naloxone and BIIE0246 act as inverse agonists at µ opioid and NPY Y2 receptors (*Elbrønd-Bek et al., 2015*; *Wang et al., 2001*) and consequently inhibit constitutive GPCR activity as well as inhibiting agonist evoked responses. As these ligands augmented TRPM3 activity in DRG neurons in vitro, we examined if they would increase the nociceptive effects of TRPM3 agonists. For these studies we used intraplantar injection of PS alone, which evoked relatively mild behavioural responses.

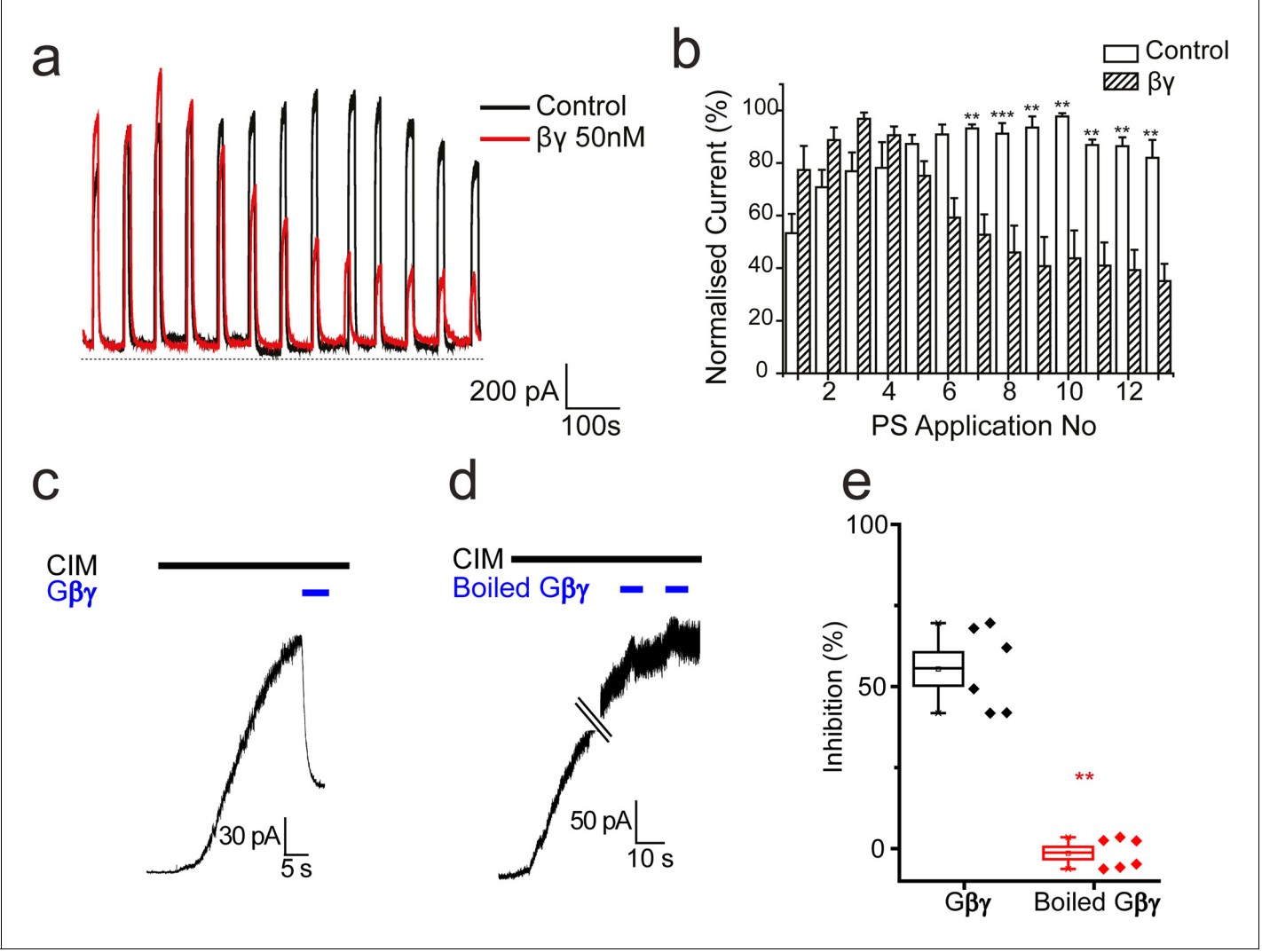

**Figure 8.** Effect of βγ subunits on PS-evoked currents. (**a**) Outward currents (+40 mV) evoked by 10 s PS (50 µM) applications every minute in TRPM3-expressing HEK293 cells. Cells dialysed intracellularly with βγ subunits (50 nM, red trace) showed a progressive decline in the evoked current amplitude compared to control cells. (**b**) Column graph displaying normalised current (% of maximum current) for PS-evoked outward currents in TRPM3-expressing HEK293 cells. Columns represent mean ± SEM; **p<0.01, ***p<0.001, unpaired t-test (control, n = 3–5 cells; βγ, n = 3–5 cells). (**c**) βγ subunits (50 nM) applied to the intracellular face of an inside out membrane patch from TRPM3-expressing HEK293 cell inhibit 10 µM CIM0216 evoked outward current (+60 mV). (**d**) Heat inactivated (100°C, 10 min) βγ subunits (50 nM) do not inhibit CIM0216 evoked currents. (**e**) Plots of percentage inhibition of CIM0216 evoked currents in inside out membrane patches for βγ subunits (n = 5) and boiled βγ subunits (n = 6).** p=0.002

Prior intraperitoneal administration of either 2.5 mg/kg naloxone (*Figure 9d*) or 3 mg/kg BIIE0246 (*Figure 9e*) resulted in significantly increased nociceptive responses to PS.

## Discussion

TRPM3 is expressed in DRG neurons where it plays a role in the transduction of thermal (heat) stimuli in normal conditions and notably in the development of heat hypersensitivity in inflammatory conditions (*Vriens et al., 2011*). Studies of TRPM3 have been facilitated by the discovery of PS and the synthetic compound CIM0216 as TRPM3 agonists (*Wagner et al., 2008*; *Held et al., 2015*) that can be used to probe the functions and characteristics of TRPM3, and we have utilised these compounds to investigate the regulation of TRPM3 channels by GPCR ligands. Like other TRP channels, TRPM3 activity is regulated by intracellular PI(4,5)P$_2$ and other phosphoinositides (*Badheka et al., 2015*;

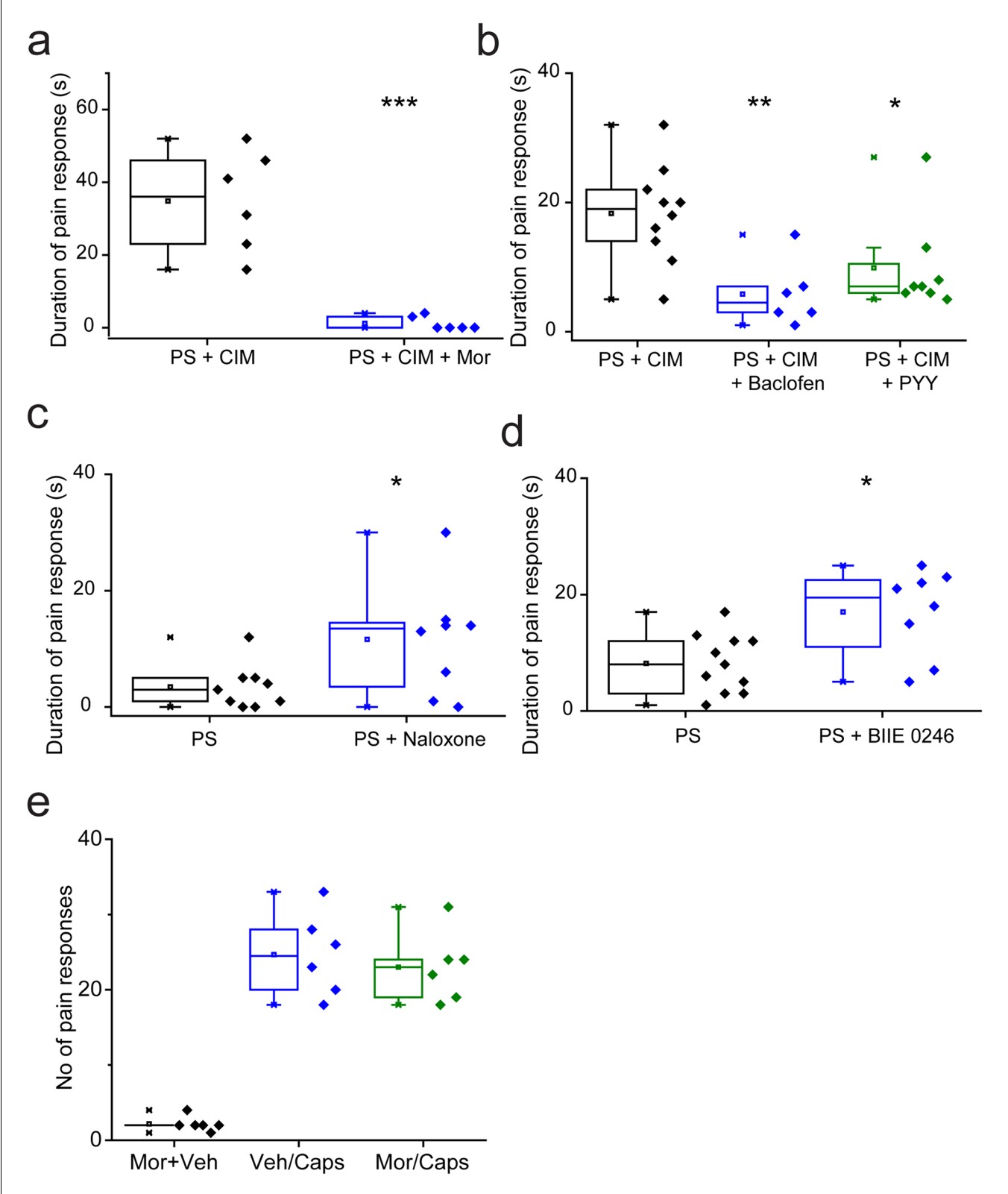

**Figure 9.** Nociceptive responses to TRPM3 agonists are modulated by Gi GPCR ligands. (**a,b**) Inhibitory effects of prior (5 min) intraplantar administrations of (**a**) 130 nmole morphine (n = 6 for each group) and (**b**) 240 nmole baclofen (n = 6) or 235 pmole PYY (n = 8) on the duration of licking/flinching evoked by intraplantar injection of 5 nmole PS plus 0.5 nmole CIM-0216 (n = 10 for PS + CIM0216). (**c,d**) Effects of prior (30 min) intraperitoneal administration of either (**c**) 2.5 mg/kg naloxone (n = 9 for PS, n = 8 for PS + naloxone) or (**d**) 3 mg/kg BIIE 0246 (n = 11 for PS, n = 8 for PS + BIIE 0246)

*Figure 9 continued on next page*

*Figure 9 continued*

on licking/flinching behaviour evoked by intraplantar administration of 5nmole PS. *p<0.05, **p<0.01, ***p<0.001; ANOVA followed by Tukey's HSD test. (**e**) No effect of intraplantar morphine (130 nmole) on nociceptive responses evoked by intraplantar administration of capsaicin (33 nmole). Vehicle, 0.9% NaCl.

*Tóth et al., 2015*) but otherwise little is known about the mechanisms that regulate the activity of this channel. Other sensory neuron TRP channels, notably TRPV1 and TRPA1, are regulated downstream of GPCR signalling by mechanisms that involve either $G_q$ activation, $PI(4,5)P_2$ hydrolysis and protein kinase C or $G_s$ activation leading to protein kinase A mediated phosphorylation. These and other phosphorylation pathways are important regulatory mechanisms for TRP channels particularly in inflammatory conditions (see *Veldhuis et al., 2015* for review). Activation of the $G_i$ coupled µ opioid receptor is a potent analgesic mechanism in sensory systems and opioid agonists such as morphine can inhibit TRPV1 activity by reducing phosphorylation levels; however, this is evident only when TRPV1 is sensitized via the cAMP-PKA pathway (*Endres-Becker et al., 2007*; *Vetter et al., 2006*, *2008*). This latter observation is consistent with our observation that morphine has little or no inhibitory effect on capsaicin evoked $Ca^{2+}$ responses when desensitization is blocked by cyclosporin.

Our results demonstrate for the first time that agonists acting at several $G_i$ coupled GPCRs (µ opioid, GABA-B and NPY receptors) exert an inhibitory effect on TRPM3 activation in DRG neurons. This inhibitory effect was PTX-sensitive demonstrating a $G_{i/o}$ protein involvement. The inhibition was not, however, mediated by the canonical $G\alpha$ subunit/adenylate cyclase pathway as the inhibitory effect was not abrogated by either application of the $G\alpha$ subunit inhibitor NF023 or by providing a membrane permeable cAMP analogue that is well known to activate PKA. Instead, our results indicate that the TRPM3 inhibition is mediated by $G\beta\gamma$ subunits. The $G\alpha_i$ inhibitor NF023, which inhibits $G\alpha_i$ subunit interactions with effector molecules, was without effect when dialysed into cells at 100 µM which is higher than its $IC_{50}$ value (~300–400 nM). However, $G\alpha$ subunits bind to effector molecules with picomolar to nanomolar affinities and the effectiveness of NF023 will depend on its relative binding affinity for $G\alpha_i$ compared to that of the effector molecules. We therefore cannot be certain that NF023 effectively inhibits $G\alpha_i$ signalling even at the concentration used. In contrast, we have clear evidence for a role for $G\beta\gamma$ subunits.

The inhibitory effects of morphine on TRPM3 were reversed by gallein, which is generally considered to be a specific inhibitor of $G\beta\gamma$ signalling (*Lin and Smrcka, 2011*), although we cannot exclude a non-$G\beta\gamma$ 'off target' effect. Critically, we found that PS evoked currents were inhibited by direct application of purified $G\beta\gamma$ subunits to either whole cells or excised inside-out membrane patches, without concomitant activation of $G\alpha_i$ or G-protein-coupled receptors. The effect of the $G\beta\gamma$ subunits was lost after heat inactivation indicating that proteins in the sample were responsible for the inhibitory activity. Our findings are therefore consistent with activation of a µ opioid receptor and an action of $G\beta\gamma$ subunits to inhibit TRPM3. A direct interaction between the released $G\beta\gamma$ subunits and TRPM3 is likely as found for some other ion channels (*Elbrønd-Bek et al., 2015*; *Wang et al., 2001*; *Veldhuis et al., 2015*). $G\beta\gamma$ subunits can, however, activate other molecules such as phospholipase C (*Rebecchi and Pentyala, 2000*), and such an action would hydrolyse $PI(4,5)P_2$ and reduce TRPM3 activity (*Badheka et al., 2015*; *Tóth et al., 2015*). Inhibition of TRPM3 activity could therefore be due to a $G\beta\gamma$/PLCmediated loss of $PI(4,5)P_2$. $PI(4,5)P_2$ levels are not maintained in isolated membrane patches and decline rapidly especially in the absence of Mg-ATP, which is required for phosphoinositide kinase mediated generation of $PI(4,5)P_2$ (*Zakharian et al., 2011*). This loss of $PI(4,5)P_2$, accounts for the greatly reduced TRPM3 channel activity previously reported in excised membrane patches (*Badheka et al., 2015*; *Tóth et al., 2015*). In our excised inside-out patch experiments, recordings were made minutes after excision into Mg-ATP free solution, which will deplete $PI(4,5)P_2$, and no further run down of channel activity was noted during the recordings. The finding that $G\beta\gamma$ subunits exerted a robust inhibitory effect on TRPM3 currents in membrane patches in these conditions suggests that the $G\beta\gamma$ inhibition does not involve PLC mediated hydrolysis of $PI(4,5)P_2$. Such a conclusion is further supported by the observations in DRG $Ca^{2+}$ imaging experiments where, in contrast to TRPM3, TRPV1 responses were not affected by morphine. As the activities of both channels are modulated by $PI(4,5)P_2$ hydrolysis (*Badheka et al., 2015*; *Cao et al., 2013*),

the differential, strong inhibition of TRPM3 cannot be explained by PLC activation and a reduction in PI(4,5)P$_2$.

Direct effects of G protein subunits on ion channel functions have been well studied for P/Q- and N-type voltage gated calcium channels (*Dolphin, 2003*) and Kir channels (*Dascal, 1997*; *Yamada et al., 1998*). In contrast, there is relatively little knowledge of direct G protein subunit - TRP channel interactions. TRPM1 channels are inhibited by activation of G$_o$–linked GPCRs and the available evidence is consistent with regulation by interactions direct between TRPM1 and both G$\alpha o$ and G$\beta\gamma$ subunits (*Shen et al., 2012*; *Xu et al., 2016*; *Devi et al., 2013*; *Koike et al., 2010*). In other studies, TRPM8 was shown to be inhibited by an interaction with G$\alpha_q$ (*Zhang et al., 2012*; *Li and Zhang, 2013*) while TRPA1 activation following stimulation of MrgprA3 receptors in DRG neurons was inhibited by gallein and phosducin, consistent with a G$\beta\gamma$ mediated mechanism (*Wilson et al., 2011*). TRP channel regulation by direct G protein subunit interactions is therefore emerging as an important and novel mechanistic concept.

DRG neurons showed differential sensitivity to activation of different GPCRs. The TRPM3 responses in many neurons were inhibited by more than one GPCR agonist consistent with expression of multiple GPCRs in these neurons. Presumably the inhibition pattern reflects the distribution and expression levels of the different GPCRs. However, not all G$_{i/o}$ GPCR agonists inhibited TRPM3. The group III mGluR agonist, L-AP4, and agonists at δ- and κ-opioid receptors did not significantly inhibit TRPM3 responses although there is good evidence for their expression in DRG neurons (*Scherrer et al., 2009*; *Carlton and Hargett, 2007*; *Govea et al., 2012*; *Honsek et al., 2015*; *Wang et al., 2010*). In part this could be explained by lack of TRPM3-GPCR co-expression in individual DRG neurons. For example, δ-opioid receptors are expressed by about 15% of, mainly larger diameter, DRG neurons (*Scherrer et al., 2009*; *Bardoni et al., 2014*) although there is evidence for some co-expression of μ- and δ-opioid receptors (*Devi et al., 2013*), There is also likely to be an overlap in expression of G$_{i/o}$ –coupled metabotropic glutamate GPCRs (*Carlton and Hargett, 2007*; *Govea et al., 2012*) with the small-medium diameter DRG neurons that express TRPM3 (*Vriens et al., 2011*). A significant inhibition of TRPM3 responses would therefore be expected in some DRG neurons, as seen for CB1 receptor activation (*Agarwal et al., 2007*; *Veress et al., 2013*). The effect of CB1 receptor activation in DRG neurons was restricted to a small number of neurons, although our experiments with heterologously expressed TRPM3 and CB1 indicated that CB1 receptor activity can regulate TRPM3. It is possible that specific macromolecular assemblies are required for the efficient interaction between types of GPCRs and TRPM3 and that δ- and κ-opioid and mGluR receptors do not contribute to these complexes, and that CB1 receptors are variably coupled in DRG neurons. The pronounced inhibition of TRPM3 by μ-opioid receptor activation is interesting as this receptor sub-type is expressed in mice in heat sensitive DRG neurons (see *Scherrer et al., 2009*; *Honsek et al., 2015*) and may be functionally relevant for opioid control of heat sensation.

An interesting additional finding was that while both baclofen and PYY robustly inhibited TRPM3 responses in a substantial percentage of DRG neurons, this inhibitory effect was only partially reversed by gallein. In contrast, gallein strongly reversed the inhibitory effect of morphine on TRPM3. This finding raises the possibility that G$_{i/o}$-coupled GPCRs can regulate TRPM3 via several signalling pathways.

In addition to demonstrating GPCR agonist induced inhibition of TRPM3, our in vitro studies have revealed that inverse agonists that act at μ-opioid and NPY receptors (naloxone and BIIE0246) can potentiate TRPM3 mediated responses. These findings are consistent with the concept that constitutive activity of μ-opioid and NPY receptors provides a level of tonic inhibition of TRPM3. A potentiating effect of these ligands was also noted in vivo, where they potentiated the behavioural effects of local intraplantar injection of PS. This result could reflect an inhibition of constitutive GPCR activity, as suggested by the in vitro findings, or inhibition of endogenous GPCR agonists produced in the tissues. Intraplantar administration of morphine, baclofen or PYY inhibited the strong nociceptive behavioural responses evoked by combined local application of PS and CIM0216. Such a peripheral anti-nociceptive action could be due to an action that inhibits action potential transmission, perhaps by activation of voltage gated potassium channels. However, morphine did not inhibit TRPV1-mediated, capsaicin-evoked pain responses so a general inhibitory action of morphine on the transmission of nociceptive signals can be ruled out. The inhibition of TRPM3-mediated nociceptive responses by the GPCR agonists can therefore be correlated to TRPM3 inhibition. Our results demonstrate that GPCR modulation of TRPM3 occurs in vivo and that the effects are localized to the regions of

sensory nerve terminals rather than systemic effects operating at the level of the spinal cord or higher centres in the nociception pathway.

Our results demonstrate that TRPM3 in sensory neurons is subject to $G_{i/o}$ GPCR regulation via a $G\beta\gamma$ subunit action. Activation of μ-opioid and GABA-B receptors are important analgesic mechanisms that operate both peripherally and centrally, including actions on the central terminals of sensory nerves in the spinal dorsal horn. Given the co-expression of TRPM3 and these GPCRs in nociceptive sensory neurons and the emerging role of TRPM3 in nociception, our findings highlight the importance of determining the role of TRPM3 in pathophysiological pain conditions.

## Materials and methods

### Cell culture

DRG neurons were prepared from adult male or female C57Bl/6J mice using methods described previously[48]. Isolated neurons were plated on poly-d-lysine coated coverslips and maintained at 37°C in an atmosphere of 95% air-5% $CO_2$ in MEM AQ (Sigma, Poole, UK) supplemented with 10% fetal bovine serum (FBS), 100 U/ml penicillin, 100 μg/ml streptomycin, and 50 ng/ml NGF (Promega, Southampton, UK) for up to 24 hr before experimentation. HEK293 cells (RRID:CVCL_U427, Thermo-Fisher Scientific) stably expressing TRPM3α2 plasmid (pcDNA3.1) DNA (provided by Dr Stephan Philipp; University of Saarland, Homburg, Germany) were grown in DMEM AQ supplemented with penicillin (100 U/ml), streptomycin (100 μg/ml), FBS (10%) and G418 (0.5 mg/ml). CHO cells (RRID: CVCL_U424, ThermoFisher Scientific) stably expressing TRPM3α2 were grown in MEM AQ media with the same supplements noted for HEK293 cell growth. For some experiments, TRPM3 HEK293 or CHO cells were transiently transfected with plasmids encoding mouse μ opioid receptor-IRES-GFP (pCMV6, NM_001039652) or GFP tagged-rat CB1 receptor (pEGFP-N1, NM_012784.4), using Lipofectamine 2000 according to the supplier's protocol. All cells used were mycoplasma free.

### Imaging changes in intracellular calcium levels

DRG neurons were loaded with 2.5 μM Fura-2 AM (Molecular Probes) in the presence of 1 mM probenecid for ~1 hr. Dye loading and all experiments were performed in a physiological saline solution containing (in mM) 140 NaCl, 5 KCl, 10 glucose, 10 HEPES, 2 $CaCl_2$, and 1 $MgCl_2$, buffered to pH 7.4 (NaOH). Drug solutions were applied to cells by local microsuperfusion of solution through a fine tube placed very close to the cells being studied. The temperature of the superfused solution (24–25°C) was regulated by a temperature controller (Marlow Industries) attached to a Peltier device with the temperature measured at the orifice of the inflow tube. Images of a group of cells were captured every 2 s at 340 and 380 nm excitation wavelengths with emission measured at 520 nm with a microscope based imaging system (PTI, New Jersey, USA). Analyses of emission intensity ratios at 340 nm/380 nm excitation (R, in individual cells) were performed with the ImageMaster suite of software.

### Electrophysiology

DRG neurons and TRPM3 expressing HEK293 or CHO cell lines were studied under voltage-clamp conditions using an Axopatch 200B amplifier and pClamp 10.0 software (RRID:SCR_011323, Molecular Devices). DRG neuron recordings were performed at +40 or+60 mV using borosilicate electrodes (3–6 MΩ) filled with (in mM): 140 CsCl, 10 EGTA, 1 $CaCl_2$, 2MgATP, 2 $Na_2ATP$, buffered to pH 7.4 (CsOH). DRG neurons were studied in the extracellular solution used for $[Ca^{2+}]_i$-measurements (see above). HEK293 and CHO cells were studied under Ca2+-free conditions (same solution as above, but with $CaCl_2$ substituted by 1 mM EGTA). Excised inside-out membrane patch recordings were made using TRPM3 expressing HEK293 cells. The solution bathing the intracellular membrane face of excised membrane patches contained either (mM) 140 CsCl, 10 EGTA, 10 HEPES or 70 CsCl, 70NMDG-Cl, 10 EGTA, 10 HEPES pH7.3. For some experiments current-voltage relationships were obtained from voltage ramps (300 ms duration) from −100 mV to voltages up to +200 mV. The NMDG containing intracellular solution was used for voltage ramp studies to reduce the amplitude of the outward current.

$\beta\gamma$ subunits from bovine brain (Millipore Ltd., Watford, UK) were dissolved in intracellular solution at a final concentration of 50 nM. For some experiment the $\beta\gamma$ subunits were inactivated by heating

to 100°C for 10 min. $\beta\gamma$ subunits (plus CIM0216) were applied by pressure ejection from a blunt pipette positioned close to the excised membrane patch.

## Behavioural assessment of pain responses

All behavioural experiments were approved by the King's College London Animal Welfare and Ethical Review Board and conducted under the UK Home Office Project Licence PPL 70/7510. PS, or a combination of PS/CIM0216 (25 µl) and morphine, baclofen and PYY (10 µl) were injected subcutaneously into the plantar surface (intraplantar, i.pl.) of one of the hind paws using a 50 µl Luer-syringe (Hamilton Co.) fitted with a 26-gauge x 3/8 inch needle. Morphine, baclofen and PYY were injected 5 min before PS/CIM0216. Naloxone (2.5 mg/kg) and BIIE 0246 3 mg/kg were injected intraperitoneally 30 min prior to intraplantar injection of PS. Mice were habituated to the experimental Perspex chambers before the experiment and placed in the chambers immediately after injection of TRPM3 agonists. The duration of pain-related behaviours (licking and biting or flinching and shaking of the injected paw) was recorded using a digital stop-watch. Total pain response times over the first 2 min were used for analysis as the pain behaviours were largely restricted to this period. Groups of 6–11 animals were used for each agent. Pregnenolone sulphate (PS,): a 200 µM solution in PBS was prepared from a 60 mM DMSO stock solution. 25 µl was injected i.pl. into the left hind paw. CIM0216: a 20 µM solution was prepared from a 5 mM DMSO stock solution. A combined PS/CIM solution was prepared by combining 1 ml 400 µM PS with 1 ml 40 µM CIM and 25 µl was injected i.pl. into the left hind paw. Naloxone HCl was administered at a dose of 2.5 mg/kg i.p. 30 min prior to PS. Morphine, 100 µg in 10 µl was injected i.pl. 5 min prior to CIM/PREGS. Baclofen, 60 µg in 20 µl was injected i.pl. 5 min prior to CIM/PS. PYY, 1 µg in 10 µl was injected i.pl. 5 min prior to CIM/PS. BIIE 0246 was made up in saline from a 10 mM DMSO stock and dosed at 3 mg/kg i.p. 60 min prior to PS.

## Materials

All compounds were from Sigma-Aldrich, Poole, UK unless otherwise stated. Stock solutions of pregnenolone sulphate, CIM0216 (Tocris Bioscience; Bristol, UK), WIN 55212–2 (Tocris Bioscience; Bristol, UK), AM251 (Tocris Bioscience; Bristol, UK), Gallein (Santa Cruz Biotechnology; Heidelberg, Germany) and BIIE 0246 (Tocris Bioscience; Bristol, UK) were made in DMSO (Calbiochem; Darmstadt, Germany). Stock solutions of morphine and naloxone were made in $H_2O$ and DAMGO (Tocris Bioscience; Bristol, UK), SB205607 (Tocris Bioscience; Bristol, UK), U50488 (Tocris Bioscience; Bristol, UK) and PYY (ABGent; San Diego, CA), were made in physiological extracellular solution. A stock solution of 8-bromo cAMP was made in $H_2O$ titrated with NaOH. Stock solutions of L-AP4 (Tocris Bioscience; Bristol, UK) and (RS)-Baclofen (Tocris Bioscience; Bristol, UK) were made in molar equivalent solutions of NaOH. Stock solutions were aliquoted and stored at −20°C. Stock solutions were diluted in physiological extracellular solution for their use in experiments. PTX (lyophilised powder) was reconstituted in $H_2O$ and was stored at 4°C. PTX was added to cells at a concentration of 200 ng/ml.

## Statistical analysis

Data are presented as box and whisker plots showing the mean (square symbol), median (horizontal line), interquartile range (box) and 5% and 95% percentile points (whiskers). For the microfluorimetry and electrophysiology experiments, 'n' values represent the number of PS responding neurons or cells studied except where indicated otherwise in the text. For behavioural experiments, 'n' values represent the number of animals in each group. No statistical methods were used to predetermine sample sizes, however our sample sizes are similar to, or greater than, those generally employed in other studies in the field. Normality of data was tested using the Shapiro-Wilk Test. Differences in normally distributed data means between two groups were analysed using an independent samples t-test. Differences in normally distributed data means between three groups or more were analysed using a one-way ANOVA, followed by a Tukey's HSD post-hoc test. Differences in non-normally distributed data means between two groups were analysed using a Mann-Whitney U test. Differences in non-normally distributed data means between three groups or more were analysed using a Kruskal-Wallis test. All statistical analyses were made using IBM SPSS statistics, version 22 (RRID:SCR_002865).

## Acknowledgements

These studies were supported by MRC Project Grant MR/L010747/1. We thank Prof. Stephan Philipp for providing a plasmid encoding mouse TRPM3α2.

## Additional information

### Funding

| Funder | Grant reference number | Author |
|---|---|---|
| Medical Research Council | MR/L010747/1 | David A Andersson<br>Stuart Bevan |

The funders had no role in study design, data collection and interpretation, or the decision to submit the work for publication.

### Author contributions

TQ, Conceptualization, Investigation, Writing—original draft, Writing—review and editing; OA, Investigation; CG, Conceptualization, Investigation; DAA, SB, Conceptualization, Funding acquisition, Investigation, Writing—original draft, Writing—review and editing

### Author ORCIDs

Talisia Quallo, http://orcid.org/0000-0002-1998-5597
Clive Gentry, http://orcid.org/0000-0002-1148-8203
David A Andersson, http://orcid.org/0000-0001-7451-8548
Stuart Bevan, http://orcid.org/0000-0002-8977-1797

### Ethics

Animal experimentation: All animal studies were carried out according to the UK Home Office Animal Procedures (1986) Act and were approved by the King's College London Animal Welfare and Ethical Review Board (UK Home Office PPL 70/7510). Mice were killed by cervical dislocation.

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
