## [Decision Letter]

Thank you for submitting your article "Gβγ mediate Gi/o coupled GPCR inhibition of TRPM3 in sensory neurons" for consideration by *eLife*. Your article has been reviewed by three peer reviewers, and the evaluation has been overseen by Kenton Swartz as the Reviewing Editor and Richard Aldrich as the Senior Editor. The following individuals involved in review of your submission have agreed to reveal their identity: Thomas Voets (Reviewer #1); László Csanády (Reviewer #2); and Alexander Chesler (Reviewer #3).

The reviewers have discussed the reviews with one another and the Reviewing Editor has drafted this decision to help you prepare a revised submission.

Summary:

This manuscript is one of three that reports exciting new findings on the mechanism of inhibition of TRPM3 channel activity in dorsal root ganglion (DRG) sensory neurons by stimulation of G protein-coupled receptors (GPCRs). Although all three manuscripts received favorable reviews, the essential revisions will require different amounts of time to address, and we encourage the authors to coordinate submission of their revised manuscripts.

The general conclusion of this manuscript is that TRPM3 channel currents are directly inhibited by G-β-γ subunits released upon stimulation of various GPCRs coupled to heterotrimeric Gi/o proteins, and this inhibition is independent of any effects on cAMP concentrations. The majority of experiments reflect Ca^2+^ imaging on dissociated dorsal root ganglion (DRG) neurons, except for a few whole-cell patch-clamp recordings, and some behavioral experiments in live animals. This study was less mechanistically oriented than the companion submissions by two other groups, and some conclusions rely on faith in generally accepted phenomena. These include the potential effects on PIP2 levels that are not directly addressed (the assumption is that those remain unaffected by the Gi/o pathway), and the involvement of G-α is excluded simply based on the lack of effect of a known G-α inhibitor. On the other hand, the results of this study complement those of the co-submissions in several ways:

i) TRPM3 inhibition through a different subset of GPCRs present in DRG neurons is investigated here. Stimulation of opioid μ,GABA-B, neuropeptide Y, and cannabinoid CB1 receptors are shown to inhibit TRPM3 currents in subpopulations of DRG neurons.

ii) Stimulation of submaximally activated TRPM3 channels by inverse agonists of either opioid μ or NPY receptors suggests some tonic inhibition of TRPM3 currents by constitutive activity of these GPCRs (or perhaps the presence of low levels of endogenous ligands). The relevance of this tonic inhibition is also demonstrated through in vivo experiments that show increased sensitivity to TRPM3 agonist-induced pain in the presence of opioid μ or NPY receptor blockers.

Essential revisions:

1) What is missing in this manuscript is an indication of the selectivity of the receptor activation towards TRPM3-dependent pain signals. To correlate the reduced pain response in the presence of morphine or baclofen to the inhibition of TRPM3 function, it is essential to show that other TRPM3-independent pain responses are not affected. This would involve demonstration that neuronal calcium signals and pain evoked by e.g. capsaicin or mechanical stimuli are not affected by these agonists.

2) This study would be greatly strengthened if the authors showed that activation of one of their GPCRs could inhibit TRPM3 currents in HEK cells. It would also be good to demonstrate that G-β/γ applied to inside out patches can inhibit TRPM3 to strengthen the claim of a direct interaction with the channel. It is also not clear what the control condition is for Figure 5, but ideally it would be boiled GBG protein.

3) To fully evaluate the quality and robustness of calcium imaging data it is helpful to have representative images as well as traces. This would allow the reader to appreciate the density and health of the cells and neurons as well as the robustness of the responses. Also, glia vs. neurons in the culture should not be highlighted. Absence of KCl response is a poor proxy for glial identification – if this is the only criteria, then calling those cells KCl-negative cells is more appropriate.

4) The scatter of data used to generate bar graphs is unclear. These should be included on all summary plots throughout the manuscript to provide a better sense of the effect size and variability. The number of tested cells is quite low for the experiments in Figure 3, so it is unclear how significant and representative the derived percentages are. The authors should also provide information on how many neurons responded to PS and what percentage of them are inhibited by Gi/o agonists?

5) There are no positive controls in Figure 4. Since these are all negative findings, it is difficult for the reader to appreciate how well the pharmacological manipulations are working.

6) Gallein is known to inhibit other targets, the short comings of this drug should at least be discussed, and it would be ideal if orthogonal approaches to interfere with GBG signaling were added to strengthen the findings (such as PTX or BARK-ct).

---

## [Author Response]

*Essential revisions:*

*1) What is missing in this manuscript is an indication of the selectivity of the receptor activation towards TRPM3-dependent pain signals. To correlate the reduced pain response in the presence of morphine or baclofen to the inhibition of TRPM3 function, it is essential to show that other TRPM3-independent pain responses are not affected. This would involve demonstration that neuronal calcium signals and pain evoked by e.g. capsaicin or mechanical stimuli are not affected by these agonists.*

We have added experiments examining the effects of morphine on capsaicin-evoked Ca^2+^ responses in DRG neurons and capsaicin-evoked pain responses in vivo. For the former experiments we used conditions where TRPV1 desensitization was greatly reduced by the presence of cyclosporin to inhibit calcineurin. Under these conditions, morphine did not inhibit TRPV1 (new Figure 5). In vivo, morphine did not affect the nocifensive, behavioural responses (licking/flinching) evoked by local intraplantar injection of capsaicin (new Figure 9). The inhibitory effect of morphine on licking/flinching evoked by the TRPM3 agonists therefore correlates with inhibition of TRPM3 activity rather than a general inhibition of sensory neuron responses.

*2) This study would be greatly strengthened if the authors showed that activation of one of their GPCRs could inhibit TRPM3 currents in HEK cells. It would also be good to demonstrate that G-β/γ applied to inside out patches can inhibit TRPM3 to strengthen the claim of a direct interaction with the channel. It is also not clear what the control condition is for Figure 5, but ideally it would be boiled GBG protein.*

We have added the results of experiments where we applied G-β/γ subunits directly to inside-out membrane patches from TRPM3 expressing HEK293 cells. These results complement the whole cell experiments in the original submission. G-β/γ subunits inhibited CIM0216 currents in the excised membrane patches by ~50% similar to the inhibition noted in whole cell experiments. As a control we used heat inactivated (boiled) G-β/γ subunits as suggested. The latter had no inhibitory effect on the CIM0216 evoked currents in inside-out membrane patches.

*3) To fully evaluate the quality and robustness of calcium imaging data it is helpful to have representative images as well as traces. This would allow the reader to appreciate the density and health of the cells and neurons as well as the robustness of the responses. Also, glia vs. neurons in the culture should not be highlighted. Absence of KCl response is a poor proxy for glial identification – if this is the only criteria, then calling those cells KCl^-^negative cells is more appropriate.*

We have added representative images of the cells and Fura-2 fluorescence signals in Figure 1 and Figure 2 to provide the requested information on the density of cells in the preparation. The term glia was not used in the original manuscript and we made no comment about non-neuronal cells. We do not believe that there are any non-neuronal cells in our preparation that will respond to a high K^+^ solution with a robust increase in intracellular calcium concentration. However, we have replaced the term ‘neuron’ with ‘cells’ in many places as a positive response to the reviewers’ comment.

*4) The scatter of data used to generate bar graphs is unclear. These should be included on all summary plots throughout the manuscript to provide a better sense of the effect size and variability. The number of tested cells is quite low for the experiments in Figure 3, so it is unclear how significant and representative the derived percentages are. The authors should also provide information on how many neurons responded to PS and what percentage of them are inhibited by Gi/o agonists?*

We have replaced the bar graphs with box and whisker plots and show all data points for all experiments. Data from additional experiments have now been added to the original Figure 3 data set increasing the number of PS-/KCl-responsive cells to 210. We believe that this gives a more accurate account of the percentages of cells affected by the various agonists. The results are also now summarized in Venn diagram (Figure 3).

*5) There are no positive controls in Figure 4. Since these are all negative findings, it is difficult for the reader to appreciate how well the pharmacological manipulations are working.*

The ability of 8-bromo-cAMP to stimulate PKA has been well established in numerous published experiments and one of the authors has published showing that it affects histamine evoked DRG Ca^2+^ responses at the concentration used in the current experiment (Nicolson TA, Foster AF, Bevan S, Richards CD. 2007 Prostaglandin E2 sensitizes primary sensory neurons to histamine. Neuroscience. 150:22-30). We have not therefore added experimental data on the ability of this compound to stimulate cAMP pathways.

We acknowledge that the action of NF023 in experiments in the original submission was questionable as the compound was applied to intact cells in calcium imaging experiments. Although the compound has been reported to affect intact cells, we believe that it will not cross the plasma membrane easily. We have therefore now included experiments where it is included in the pipette solution at a high concentration (100µM) in whole cell recordings experiments on TRPM3-expressing CHO cells (new Figure 6). This concentration of NF023 is higher than its reported IC50 value (300-400nM). The compound had no effect on morphine inhibition. In the absence of a direct positive control for the activity of the compound we have added a caveat about interpretation of the negative result and state in the Discussion ‘We therefore cannot be certain that NF023 effectively inhibits Gαi signalling even at the concentration used.’

6) Gallein is known to inhibit other targets, the short comings of this drug should at least be discussed, and it would be ideal if orthogonal approaches to interfere with GBG signaling were added to strengthen the findings (such as PTX or BARK-ct).

Experiments showing that PTX inhibits the effect of morphine were included in the study. We have made the wording slightly more prominent to emphasize this.

We have also added a caveat that gallein could have ‘off target’ non-G-β/γ subunit effects. This comment could probably be applied to almost any inhibitor used. We could not find published data that gallein inhibited other targets other than those where the effects were interpreted as being via a G-β/γ pathway. We would appreciate any information from the reviewers on this point.